# Nano-Additive Manufacturing and Non-Destructive Testing of Nanocomposites

**DOI:** 10.3390/nano13202741

**Published:** 2023-10-10

**Authors:** Yulong She, Jie Tang, Chaoyang Wang, Zhicheng Wang, Zhengren Huang, Yong Yang

**Affiliations:** 1State Key Laboratory of High Performance Ceramics and Superfine Microstructures, Shanghai Institute of Ceramics, Chinese Academy of Sciences, Shanghai 200050, China; sheyulong22@mails.ucas.ac.cn (Y.S.); 18952230772@163.com (J.T.); wangchaoyang21@mails.ucas.ac.cn (C.W.); wzc99994@163.com (Z.W.); 2College of Materials Science and Opto-Electronic Technology, University of Chinese Academy of Sciences, Beijing 100049, China

**Keywords:** additive manufacturing, nanocomposite, vat polymerization non-destructive testing, computed tomography (XCT), defect

## Abstract

In the present work, the recent advancements in additive manufacturing (AM) techniques for fabricating nanocomposite parts with complex shaped structures are explained, along with defect non-destructive testing (NDT) methods. A brief overview of the AM processes for nanocomposites is presented, grouped by the type of feedstock used in each technology. This work also reviews the defects in nanocomposites that can affect the quality of the final product. Additionally, a detailed description of X-CT, ultrasonic phased array technology, and infrared thermography is provided, highlighting their potential application in non-destructive inspection of nanocomposites in the future. Lastly, it concludes by offering recommendations for the development of NDT methods specifically tailored for nanocomposites, emphasizing the need to utilize NDT methods for optimizing nano-additive manufacturing process parameters, developing new NDT techniques, and enhancing the resolution of existing NDT methods.

## 1. Introduction

Additive manufacturing technology, commonly referred to as 3D printing, represents a novel approach to the direct and swift fabrication of three-dimensional objects. This process relies on a three-dimensional digital model of an object, with layers of powdered or filamentary materials being stacked incrementally [1,2,3,4,5,6]. Over the past few decades, this technology has experienced remarkable and rapid growth, capturing substantial attention. Additive manufacturing operates on the principles of discretization and accumulation mechanisms. In comparison to traditional methods such as equivalent manufacturing and subtractive manufacturing, additive manufacturing circumvents the limitations imposed by mold production or processing techniques. It effectively addresses the manufacturing challenges posed by intricate shapes and structures, substantially streamlining processing steps and reducing production timelines.

Notably, the advantages of additive manufacturing become increasingly pronounced as the complexity of a product’s structure intensifies. Consequently, it has found widespread applications in polymers, metals, and select ceramics [7,8,9,10,11,12,13,14,15,16,17]. Figure 1 showcases a collection of complex-shaped samples produced using various additive manufacturing methods. Specifically, when compared to conventional techniques, additive manufacturing boasts the following key advantages [1]: 1. Rapid Prototyping: The entire manufacturing process is streamlined into three stages: computer-aided design, near-net formation of blanks, and minimal machining. This eliminates the need for mold design and production, significantly reducing time and costs. This facilitates swift conversion for processing diverse components in small batches and allows for remarkably agile responses to structural design alterations, thereby shortening product development cycles. 2. Unrestricted Structural Complexity: Parts design and production are unshackled from the constraints of structural intricacy. The absence of molds enables direct fabrication of parts with intricate internal and closed cavities, liberating structural design from the limitations of manufacturing techniques. 3. Versatile Composite Manufacturing: Additive manufacturing enables the composite fabrication of parts using various materials. By flexibly adjusting local material composition and microstructure in accordance with part working conditions and performance demands, high-performance material components, including multi-material and gradient materials, can be directly near-net shaped. This capability surpasses the bounds of any previous material processing techniques, offering enhanced parts design, weight reduction, cost efficiency, and optimal utilization of performance potential. 4. True Digital and Intelligent Processing: The complete additive manufacturing process, encompassing parts design, geometric modeling, layering, and process planning, is executed within the digital realm. Computer control guides the actual processing, culminating in a fully digitized and intelligent manufacturing process. Consequently, governments and scholars worldwide have taken substantial interest in and invested resources into this technology. Meanwhile, with the great progress made in the production and processing of nanomaterials, the interest in additive manufacturing (AM) has been increasing [18,19]. Additive manufacturing and nanomaterials are often paired together; in some cases, nanomaterial-based inks can be extruded to form patterned parts in which the nanomaterials themselves are the main component [20,21,22,23]. However, more often than not, nanomaterials find application as fillers, where they serve as conductive or mechanical additives aimed at enhancing the properties of the end product. Nonetheless, there exists a relatively limited body of research concerning 3D printed nanocomposites. The available studies merely demonstrate the feasibility of producing and processing existing nanocomposites through 3D printing. 

In the process of nanocomposite additive manufacturing, defects can arise due to the complex thermodynamic behavior of the material under high-energy beams and the influence of various manufacturing parameters. The presence of defects can lead to the degradation of the final product’s performance and act as a bottleneck in the development of additive manufacturing.

Some common defects include porosity, delamination, cracking, warping, inconsistent material distribution, and surface roughness [24,25,26]. Porosity refers to the presence of voids or air pockets within the nanocomposite, weakening its structural integrity. Delamination occurs when there is separation or detachment between different printed layers, while cracking refers to the formation of cracks due to high thermal stresses. Warping refers to the deformation or distortion of the printed part, and inconsistent material distribution can lead to changes in properties and performance. Poor surface finish or roughness can result from improper control of printing parameters, or post-processing techniques.

These defects can arise due to factors such as improper printing parameters, inadequate material preparation, suboptimal machine calibration, and insufficient process control. Addressing these defects requires optimization of the additive manufacturing process parameters, material formulation, and post-processing techniques to ensure the production of high-quality nanocomposite additive manufacturing parts.

However, challenges and scientific issues persist in the field. The unique surface properties of nanomaterials exacerbate thermal stress challenges, interfacial complications, and dispersion and rheology issues during the additive manufacturing process. These challenges contribute to defects and cracks in the final nanocomposite products. Quantifying and characterizing defects, understanding the underlying mechanisms, and developing methods for defect control are crucial research challenges in nanocomposite additive manufacturing [27].

Non-destructive testing (NDT) technology plays a vital role in identifying and assessing defects within nanocomposites. NDT methods allow for the detection of defects without compromising the performance of the inspected objects. By employing various physical and chemical phenomena, NDT can provide valuable information about the shape, nature, size, location, orientation, distribution, and inclusion of defects [28,29]. The application of NDT in nanocomposite additive manufacturing enables qualified assessments and necessary treatments for the parts, contributing to defect control and overall quality improvement. Figure 2 shows the defect size–incidence distribution curve, and defect detection should exclude products with defect sizes larger than α_u_.

Initially, many additive manufacturing products referred to acceptance criteria borrowed from the materials they replaced, which served as benchmarks for product approval. However, an increasing amount of research has indicated that additive manufacturing materials exhibit unique characteristics and orientations that greatly differentiate them from traditional materials used in forgings or castings. Consequently, the defects arising from the manufacturing process differ in terms of size and distribution compared to conventional materials, challenging the appropriateness of employing raw material acceptance criteria. Therefore, it becomes imperative to establish an understanding of how defects impact the mechanical properties of additive manufacturing parts and to establish standardized non-destructive testing procedures for defect detection. This prerequisite ensures the safety of additive manufacturing parts and forms the basis for their practical application in engineering. This paper aims to consolidate information on the various methods of forming nanocomposite additively manufactured parts, the mechanisms behind defect formation, and the ongoing advancements in non-destructive testing methods within the additive manufacturing domain. Section 2 delves into the primary methods of additive manufacturing, while Section 3 outlines the key NDT techniques applied in additive manufacturing, offering a comprehensive analysis based on principal parameters. Section 4 outlines the anticipated future trends and challenges, and, finally, Section 5 presents the primary conclusions drawn from this study.

## 2. Nanocomposite Additive Manufacturing Technology

As shown in Figure 3, in the realm of rapid prototyping, the industry presently relies on seven primary categories of additive manufacturing technologies. These are as follows:

1. vat photopolymerization; 2. powder bed fusion; 3. materials extrusion; 4. binder jetting; 5. material jetting; 6. directed energy deposition; and 7. sheet lamination. A common thread woven through all these additive manufacturing methods is their capacity to construct three-dimensional physical objects by progressively layering material upon material. The focus of this paper will be directed towards the initial three techniques and their associated applications within the realm of nanocomposites. 

### 2.1. Vat Photopolymerization

Among the various additive manufacturing techniques, Stereolithography (SLA) stands out as the technology offering the most superior resolution. While 3D printers of alternative technologies typically exhibit resolutions within the range of 50–200 μm, SLA’s commercial printers can effortlessly attain resolutions of 20 μm or even finer [31,32]. This elevated resolution not only contributes to precision but also enables the production of intricately detailed printed products. The exceptional resolution of SLA technology is attributed to the precise spatial control of applied photons’ intensity. Notably, Old World Laboratory has recently unveiled SLA printers leveraging two-photon polymerization technology, achieving resolutions as fine as 100 nm. This innovation permits nanometer-level printing accuracy, a capability that empowers the printing of exceptionally intricate and delicate components on a sub-micron scale [33]. Consequently, the application scope of SLA expands significantly, encompassing an even broader array of possibilities.

While Stereolithography (SLA) technology indeed offers substantial advantages in terms of resolution, it is not exempt from encountering certain challenges. One such challenge pertains to the interlayer delamination observed in the process of nanocomposite, light-cured additive manufacturing. This phenomenon is primarily attributed to weak interlayer bonding. The root cause of this bonding issue predominantly stems from the scattering properties of nanoparticles present in the material. These properties exert an influence on the propagation of light through the resin system, subsequently leading to compromised curing quality. This issue is primarily linked to the scattering attributes of nanoparticles, ultimately impacting light propagation and, consequently, the overall curing efficacy [34,35]. Another limitation of SLA is that the printing process is relatively slow, this is mainly due to the discontinuous nature of the printing process. The basic layer-by-layer deposition mechanism of SLA requires that the laser scanning, platform movement, and resin filling must be carried out in separate, discontinuous steps. As a result, there are long periods of time between each step in which no actual printing occurs. This adds significantly to the print time [23,36].

The challenge of processing duration in additive manufacturing has been addressed by the Continuous Liquid Interface Production (CLIP) technology [37], which has revolutionized the process. CLIP has achieved rapid printing times, reducing the processing duration to a matter of minutes, representing a significant improvement in efficiency. Building upon the principles of SLA, CLIP utilizes a laser to initiate the photopolymerization of liquid resin but introduces a departure from the conventional segmented process.

CLIP employs continuous curing of the photosensitive resin, minimizing time gaps between successive steps and contributing to an expedited process. A key distinction of CLIP is the continual movement of the build platform, which maintains a slow and synchronized pace to match the resin’s curing speed. This ensures a seamless and continuous printing process. By integrating these advancements, CLIP has streamlined additive manufacturing, achieving remarkable time efficiency while building upon the foundational principles of SLA.

A schematic of the CLIP printer is shown in Figure 4, illustrating the continuous curing process and the synchronized movement of the build platform.

This technological advancement in CLIP has addressed the challenge of processing duration in additive manufacturing, allowing for faster and more efficient production of parts.

SLA nanocomposites have a wide range of applications in biomaterials, structural materials, electronic materials, and magnetic materials. As shown in Figure 5, in the work of Zhou et al., a bioink was prepared using arginine-glycine-aspartate-serene (RGDS) peptide and/or nano-crystalline hydroxyapatite (nHA) as a reinforcing phase added to polyethylene glycol diacrylate (PEGDA). The photosensitive bioink was then printed using a customized desktop SLA bioprinter, and cell regeneration was further enhanced by low-intensity pulsed ultrasound (LIPUS). The study demonstrated that the combined effects of the bioactive 3D printed scaffold and the LIPUS process improved cell proliferation, alkaline phosphatase activity, calcium deposition, and total protein content. Additionally, the Young’s modulus of the nHA-PEGDA and nHA-PEGDA-RGDS nanocomposites increased by 150%. This indicates that the addition of an nHA filler not only enhances the biocompatibility of PEGDA but also improves its mechanical properties [38].

In another study by Feng et al., lignin-coated cellulose nanocrystals (L-CNC) were doped into a methacrylic acid vinyl acetate (CMA) matrix, and 3D-printed nanocomposites were prepared using the SLA technique. The vinylation reaction occurred between the L-CNC and the MA matrix, enhancing their compatibility. The addition of L-CNC improved the thermal stability and mechanical properties of the 3D printed nanocomposites, even at low concentrations of 0.1% and 0.5% [39].

These studies demonstrate the potential of SLA nanocomposites in various applications. By incorporating nanofillers such as nHA and L-CNC, the biocompatibility, mechanical properties, and thermal stability of the printed materials can be significantly enhanced. This highlights the versatility and effectiveness of SLA in fabricating nanocomposites with improved properties for different applications.

As mentioned earlier regarding the drawbacks of SLA, there are still many fundamental problems in using SLA to print nanocomposites. For example, as shown in Figure 6, when printing complex ceramic parts by adding ceramic particles to light-curing resins, the presence of nanoparticles poses a problem to printing accuracy due to the unique surface properties of the nanoparticles, which cause changes in light scattering. It has been reported that the curing accuracy is related to the curing depth (*C_d_*~μm) and the curing width (*C_w_*~μm), both of which determine the printing details. The theoretical expressions for curing depth and curing width are derived from Beer–Lambert as follows [40].
$$C_{d}=D_{p}\ln\frac{E}{E_{C}}$$
$$C_{w}=F\sqrt{\ln\frac{E}{\varphi \cdot E_{C}}}$$
where *D_p_* (~μm) is the depth of penetration, *E* (~mW) is the exposure, and *E_c_* (~mW) is the critical exposure to initiate the polymerization reaction. *D_p_* is determined by the intrinsic properties of the resin composite, including the solid content of the nanoparticles, the size of the nanoparticles, and the difference in refractive indices between the nanoparticles and the liquid resin. *F* and φ are determined by the laser beam profile and the properties of the resin composite (refractive index, particle size, and loading concentration). *E_c_* depends only on the photoinitiator and the liquid monomer (liquid resin). *E* can be determined by the following equation:$$E=\frac{2\cdot p_{0}}{\pi \cdot w_{0}\cdot v_{s}}$$
where *p*_0_ (~mW) is the laser power at the surface of the resin, *w_0_* is the beam radius at e^−2^, and *v_s_* is the scanning speed. From the given equations, it is evident that *C_d_* and *C_w_* are largely influenced by the nanoparticle properties, including loading concentration, refractive index, and size, with other parameters being a function of the SLA printer. For example, when the refractive index of the nanoparticles differs significantly from that of the liquid resin, the laser light scatters significantly, resulting in insufficient *D_p_* and an attenuation of the curing depth. Increased light scattering also results in more resin curing around the laser beam. As a result, the cure width increases, which deteriorates the resolution. Therefore, suitable nanoparticles need to be selected for a particular liquid resin [41]. The wavelength of the laser used for SLA printing is another important parameter. Most lasers in SLA are supplied by UV lamps so the wavelength range is usually between 300 and 400 nm, but this can vary from one SLA printer to another. Therefore, photoinitiators should be carefully selected to match the wavelength of the light and absorb the light to cleave and produce free radicals.

### 2.2. Materials Extrusion

#### 2.2.1. FDM

S. Scott Crump invented the FDM 3D printer in the late 1980s, and Stratasys industrialized it in 1990 [42]. In the FDM molding process, a thermoplastic wire of a specific diameter is fed into a high-temperature nozzle by a drive gear. The molten wire is then extruded through the nozzle and deposited on top of the printing platform (*XY*-axis plane). After each layer of molten wire is cured, the print head moves upwards along the *z*-axis, or the printing platform moves downwards along the *z*-axis to deposit the subsequent new layer on top. To minimize the thermal shrinkage effect that can reduce interlayer adhesion, FDM printers can control the cooling rate of the molten layers by adjusting the temperature of the print platform [43]. Additionally, simplifying the modification of the FDM printer can help reduce nozzle clogging caused by the softening of the print wire. Due to its ease of operation and low equipment cost, FDM printing technology has been widely adopted by small companies and even private users. However, the FDM molding process is prone to structural defects, which can reduce the mechanical strength of the final printed product, due to factors such as print resolution, wire uniformity, surface roughness, layered structure, and interlayer adhesion [44,45]. By adjusting 3D printing process parameters and optimizing printing wire production processes, the negative effects during the printing process can be mitigated to some extent. Due to the directional nature of the FDM molding process, 3D printed parts often exhibit anisotropic mechanical properties.

FDM 3D printers typically use thermoplastics [46,47,48,49,50], such as polyetherimide, ABS, polycarbonate, polymethylmethacrylate, polybutylene terephthalate, polycaprolactone, nylon, polypropylene, polylactic acid (PLA), and their composites. Adding fillers is the most common and effective method to improve the mechanical properties of thermoplastic matrices. When the diameter of the filler particles is in the nanometer scale (~100 nm), the composites are classified as nanocomposites [51,52]. Strong molecular interactions between polymer resins and nanofillers enhance the mechanical, thermal, and physical properties of nanocomposites [53,54]. Nanofillers can take various forms, such as particles, flakes, or fibers, including carbon fibers, nanoclays, carbon nanotubes, graphene nanosheets, and glass fibers. Nanofillers not only improve the mechanical properties of nanocomposites, but also impart new physicochemical properties.

In another study by Gnanasekaran et al., carbon nanotubes (CNTs) and graphene nanoparticles (GNPs) were used as fillers to reinforce polybutylene terephthalate (PBT) materials. The results demonstrated that the 3D printed PBT/CNT nanocomposites exhibited improved mechanical and electrical conductivity properties compared to pure PBT [55].

Similarly, Wang et al. developed a PLA nanocomposite wire with a high content of nanocellulose (up to 30 wt%). The 3D printed composites showed mechanical properties that were comparable to pure PLA [56] (Figure 7).

These findings highlight the potential of incorporating nanofillers into polymer materials for 3D printing applications as they can enhance the overall performance of the printed composites.

Overall, these studies demonstrate that the choice of nanofillers and their incorporation in nanocomposites can regulate different properties, such as mechanical strength, electrical conductivity, and 3D printing properties. The specific type and content of nanofillers play crucial roles in achieving the desired properties in nanocomposites. 

#### 2.2.2. DIW

The DIW process, first reported by Cesarano and Calvert in 1997, is a material extrusion-based technique widely used in 3D printing. It is known for its simplicity, speed, and cost-effectiveness, and it can be applied to various materials including ceramics, metal alloys, polymers, and even edible materials [57,58,59,60].

In DIW, ceramic slurry with a specific viscosity is extruded under pressure through a nozzle and shaped layer by layer along a predetermined path. The layers are stacked on top of each other to create a three-dimensional shape. To achieve successful molding, high-quality ceramic slurry with a precise composition and viscosity is required. The slurry is often adjusted to have shear thinning characteristics by forming a gel through flocculation, adding binders or plasticizers, and incorporating gelling agents to control its properties. Typically, the slurry used has a high loading of ceramic particles and optimal additive content [57].

However, one of the drawbacks of DIW is that the formed samples often have poor surface quality and low dimensional accuracy [59]. This can be attributed to various factors, such as the rheological behavior of the slurry, nozzle clogging, and the drying process. Efforts are being made to overcome these challenges and improve the surface quality and dimensional accuracy of DIW-formed ceramic parts. Research is focused on optimizing the composition and viscosity of the slurry, improving the printing parameters, and developing post-processing techniques to enhance the final product.

Overall, despite the limitations in surface quality and dimensional accuracy, DIW remains a popular and versatile 3D printing technique for various materials. Ongoing research and advancements in process optimization will continue to refine the DIW process and expand its applications in ceramic and other material-based 3D printing.

### 2.3. Powder Bed Fusion

#### 2.3.1. SLS

Selective laser sintering (SLS) is a powder additive manufacturing technology that has received much attention in recent years. It enables free-form manufacturing of complex 3D parts by curing the powder material layer by layer with a laser. First, a layer of powder is deposited into the build chamber. Then the laser beam sinters or melts selected areas on the powder bed according to the cross-section data of the 3D CAD model to form a solid layer. Once the initial layer is complete, the build platform is lowered by 100–200 µm and a new layer of powder is applied to it [61,62]. The new layer is printed by the laser while being combined with the previous layer [63,64].This process is repeated until the final 3D physical part is produced.

SLS has several advantages over other molding techniques for manufacturing nanocomposite parts. Firstly, complex geometries can be manufactured without the need for support structures due to the self-supporting nature of the powder bed during processing. This allows for the production of complex designs that were previously difficult to achieve using traditional manufacturing methods. Second, SLS allows for efficient material utilization because unused powder can be recycled multiple times without significant changes in material properties. This not only reduces material waste, but also lowers production costs. However, the main drawbacks of SLS technology are poor surface quality, low dimensional accuracy, and degradation of material properties. Whereas, the incorporation of nanoparticles in SLS offers new possibilities to enhance the mechanical, thermal and electrical properties of parts. This section explores the advances, challenges, and potential applications of SLS molding using nanocomposites.

Nanocomposites used in SLS consist of a matrix material (e.g., polymer or metal) and nanoparticles. Nanomaterials commonly used in SLS include multi-walled carbon nanotubes (MWCNT), carbon nanofibers (CNF), Al_2_O_3_ nanoparticles, silica nanoparticles, clay nanoparticles, copper nanoparticles, silver nanoparticles, and titanium dioxide nanoparticles. These nanomaterials were selected based on their desired properties such as higher mechanical strength, thermal stability, flame retardancy, and electrical conductivity [8,65,66,67,68,69,70].

As shown in Figure 8, Ding et al. utilized the selective laser sintering (SLS) technique to fabricate flexible nanocomposites composed of thermoplastic polyether block amide (TPAE) and multi-walled carbon nanotubes (MWCNTs). These flexible TPAE/MWCNTs nanocomposites exhibited excellent performance in strain sensing, electrically induced shape memory effect, and electrical conductivity [71].

Xiong et al. employed both the ball-milling (BM) and ultrasonic dispersion-deposition-liquid-phase deposition methods to coat carbon nanotubes (CNTs) onto PA12 composite powders. They subsequently utilized selective laser sintering (SLS) 3D printing technology to create conductive polymer composites for electromagnetic interference (EMI) shielding. The resulting composites exhibited a good EMI shielding ability [72].

Although SLS preparation of nanocomposites has shown promising results, there are still some challenges. Achieving uniform nanoparticle dispersion and preventing particle agglomeration are essential to obtaining consistent material properties. Developing suitable powders with desired size, morphology and dispersion is also critical for successful SLS molding. In addition, further research and development is needed to address the issues of poor surface quality, low dimensional accuracy, and potential degradation of material properties [73].

Future research directions should focus on optimizing process parameters, developing novel nanocomposite formulations, and exploring post-processing techniques to improve the mechanical strength, dimensional accuracy, and overall performance of SLS-treated nanocomposite parts. Advances in powder development and characterization techniques will contribute to the successful fabrication of high-quality nanocomposite parts using SLS.

#### 2.3.2. Multi Jet Fusion

Jet fusion 3D printing (JF3D), also known as MJF (Multi Jet Fusion), is a recently developed additive manufacturing method. Similar to selective laser sintering (SLS), JF3D uses powdered polymers as the build material. However, instead of using a laser to sinter or melt the material, JF3D utilizes a fusing agent and detailing agent to bond the powders using infrared (IR) radiation [74].

The JF3D process consists of two main components: the recoating carriage and the printing/fusing carriage. The recoating carriage deposits a thin layer of build material across the build area, moving from top to bottom. The printing/fusing carriage then scans across the initial layer from left to right. It employs a heat energy source ahead of the HP Thermal Inkjet array to ensure consistent temperature throughout the printing process. As the fusing carriage moves laterally, the printheads in the array deposit a thin layer of fusing agents in precise locations to form the first layer, while the detailing agents define the specific geometry of the part. The fusing carriage then returns to its original position, moving from right to left, and supplies energy to fuse the areas where the fusing agents were applied. This layer-by-layer process continues until the entire part is formed.

One of the advantages of JF3D is its ability to control various properties within the printed part using transforming agents. These agents can modify properties such as electrical and thermal conductivity, translucency, color, and other material properties. This provides users with a high degree of flexibility, allowing them to place specific properties at desired locations in the fabricated part.

Another significant advantage of JF3D, compared to other powder bed fusion technologies like SLS, is the potential for reduced build time. The use of planar radiation instead of the laser scanning process in SLS can greatly decrease the overall fabrication time [75].

In summary, JF3D or MJF is an innovative additive manufacturing method that utilizes fusing and detailing agents with IR radiation to bond powdered polymers. They offer flexibility in controlling specific properties within the printed part and has the potential for faster build times compared to other powder bed fusion techniques.

### 2.4. Binder Jetting

The binder jetting (BJ) process, invented by Sachs et al. in 1989, is a technique used for rapid production of parts made from various materials including plastics, metals, and ceramics. The process involves depositing organic binders onto the surface of particles in a powder bed using a print head along a predetermined path to achieve shaping. One of the main advantages of this technology is its ability to produce large-sized parts, saving significant time compared to other additive manufacturing techniques [76,77].

The application of the BJ process on ceramic materials was first reported by Sachs et al. in 1992, using alumina and silicon carbide as the matrix and colloidal silica as the binder. While the BJ process can be used with any powder, it should be noted that, similar to selective laser sintering (SLS), the entire part is formed layer by layer. After each layer is solidified, a coating system is used to distribute a new layer of powder onto the previous surface, followed by the repetition of the bonding and coating cycle until the final 3D part is formed [78].

The use of nanoparticles in the BJ process can pose challenges, particularly in the powder spreading process. Nanoparticles tend to have poor flowability due to the stronger van der Waals forces between them compared to their individual weights. This can result in accumulation defects in the powder bed [76]. Overcoming these challenges and ensuring the quality of powder spreading is crucial for the further application of BJ in ceramic nanocomposite materials.

Although the BJ process still faces challenges such as poor uniformity of the final product, it is expected that in the coming years, BJ will become a widely used technology for manufacturing 3D printed parts of ceramic nanocomposites. The continuous advancements in the BJ process and the increasing understanding of nanoparticle behavior in the powder spreading process will contribute to the wider adoption of this technology.

### 2.5. Material Jetting

#### 2.5.1. Inkjet

Inkjet printing, also known as material jet technology, can be traced back to the nineteenth century when physicist Wiliam Kelvins filed a patent for the directional deflection of liquid droplets under electrostatic forces. However, it was not until the 1950s that Siemens utilized this patent to create the first inkjet printer. With advancements in manufacturing technology, inkjet printers have become more affordable and compact, and are now widely used [79,80,81].

Inkjet printing technology enables the layer-by-layer manufacturing of structures in a highly complex manner. In this process, liquid ink is ejected drop by drop from a print head onto a build plate and cured through phase change, chemical reaction, or solvent evaporation. Inkjet printing can be categorized into continuous inkjet printing (CIJ) and drop-on-demand inkjet printing (DOD). In CIJ printing, ink is continuously ejected in droplets through nozzle electrodes, and the droplets pass through an electric field that deflects them to varying degrees. Even when printing is not required, CIJ printing continues to produce a continuous stream of ink droplets, with the unwanted droplets being collected by changing the electric field. On the other hand, DOD printing uses air pressure to hold the ink at the nozzle, and a pressure pulse is applied to the ink through a piezoelectric system. If the pulse exceeds a certain threshold, a droplet is ejected. Without a pressure pulse, the ink remains in place due to surface tension. The size of the droplets can be controlled by adjusting the pressure pulse.

Nanoparticles have been extensively studied as fillers in polymer matrix composites over the past two decades [82,83]. Even a small addition of nanoparticles can increase the Young’s modulus of composites without affecting toughness. However, the main challenge in utilizing nanomaterials is their tendency to agglomerate and the difficulty in dispersion. Commercial nanoparticles often exist in an aggregated form, and even with the addition of dispersants that adsorb onto the surface of the nanoparticles, they still tend to aggregate in the matrix. As a result, it is often observed that the improvement in mechanical properties decreases after reaching the optimal nanoparticle concentration [82].

#### 2.5.2. Aerosol Jet Printing

Aerosol jet printing (AJP) is an emerging non-contact, numerically controlled thin film printing technology that has gained attention in recent years. Its working principle involves the jetting of aerosol droplets through the simultaneous injection of sheath gas to achieve high printing accuracy and excellent boundary controllability.

The basic process of aerosol jet printing is as follows: the material to be printed is dispersed or dissolved in a suitable solvent to form a stable dispersion or solution. This prepared dispersion or solution is then atomized using either ultrasonic or pneumatic atomization, resulting in the formation of numerous micro-droplets. These micro-droplets, under the influence of a carrier gas, form an aerosol beam that moves towards the nozzle. Before leaving the nozzle, a binding gas is introduced to bind the aerosol beam. The aerosol beam is then deposited onto the substrate surface with the assistance of computer control, allowing for precise patterning. It is worth noting that during the aerosol jet printing process, the dispersion of the target material is atomized into micro-droplets, which act as micro-reactors. This enables in situ control and adjustment of the material’s morphology and structure at the microscale, making it suitable for the preparation of materials with specific micro-nanostructures.

Based on the working principle of aerosol jet printing, it offers several advantages in the printing process of nanocomposites. Firstly, it provides high printing resolution (up to 10 µm [84]. Secondly, the deposition of microdroplets occurs in a confined environment, accompanied by heat and mass transfer processes, which effectively regulate the micro-nanostructures of the deposited materials. Lastly, compared to other printing technologies, the aerosol jet printing process is relatively simple and easy to operate, and the nozzle is less prone to clogging [85,86].

## 3. Non-Destructive Testing

In the process of nanocomposite additive manufacturing, defects can occur due to the complex thermodynamic behavior of the material under the high-energy beam, as well as the influence of manufacturing parameters, surface properties of the powder material, and molding temperature. And, at the same time, because the van der Waals forces between nanoparticles are greater than their respective weights, which usually results in nanoparticles that are prone to agglomeration as well as poor mobility, not only are new defects introduced when nanofillers are added, but also, existing defects are enhanced. These defects during the printing process can lead to the degradation of the performance of the final product and become a bottleneck in the development of additive manufacturing [87,88]. Some of the common defects include:Porosity: Porosity is the presence of voids or air pockets in nanocomposites. Pneumatic holes are the most common defects in the forming process of additive manufacturing, and their sizes are mostly in the range of tens of micrometers to hundreds of micrometers, and they are randomly distributed within the parts, either in a single form or densely populated with multiple pneumatic holes. These voids weaken the structural integrity of the part and affect its mechanical properties. The introduction of nanoparticles often results in an increase in localized porosity due to nanoparticle agglomeration.Delamination, cracking: Delamination occurs when there is a separation or delamination due to poor interlayer bonding or high thermal stresses between different printed layers during the printing process of additive manufacturing samples. The size of such defects is usually large, with dimensions up to the millimeter scale. Often the introduction of ceramic nanoparticles with their high light absorption and refractive properties usually leads to increased delamination and cracking.Warping: Warping is the deformation or distortion of a printed part, usually caused by uneven cooling or residual stresses within the nanocomposite. Warping can lead to dimensional inaccuracies and affect the overall functionality of the part 6.Inconsistent material distribution: Dispersion or uneven distribution of nanoparticles in nanocomposites can lead to changes in material properties and performance. This can affect the overall quality and functionality of the printed part.Surface roughness: Poor surface finish or roughness may result from improper control of printing parameters or improper post-processing techniques. Surface roughness can affect the aesthetics and functionality of the part.

These defects can arise from various factors, including improper printing parameters, inadequate material preparation, suboptimal machine calibration, and insufficient process control. It is important to address these defects through optimization of the additive manufacturing process parameters, material formulation, and post-processing techniques to ensure the production of high-quality nanocomposite additive manufacturing parts.

Table 1 presents the characteristics of different image analysis tests for additive manufacturing samples. It can be observed that certain test methods, such as Focused Ion Beam (FIB), Electron Tomography (ET), Atom Probe Tomography (APT), etc., require the samples to be cut due to the destructive treatment involved in the testing process. However, the hierarchical nature of additive manufacturing often leads to non-real structures in the samples as a result of cutting, which can affect the analysis of defects. Another method, Nuclear Magnetic Resonance Imaging (NMR), is limited in its application for additive manufacturing due to its low resolution and high cost. Therefore, there is a need for the development of new techniques to detect defects in additive manufacturing samples.

Non-destructive testing (NDT) plays a crucial role in evaluating defects without causing damage to the inspected objects. This makes it an essential method for ensuring the production of high-quality materials and components that can be used safely and reliably. NDT also contributes to quality control, savings in raw materials, process improvement, and enhanced labor productivity. It finds extensive applications in various industries, including aviation and aerospace, nuclear technology, weapons systems, power station equipment, railway and shipbuilding, petroleum and chemical industry, boilers and pressure vessels, construction, metallurgy, and machinery manufacturing [89].

NDT technology has seen rapid development with advancements in modern physics, material science, microelectronics, and computer technology. There are more than 70 different NDT methods that have been applied and studied, covering various principles and methods. These include ray detection (X-rays, γ-rays, high-energy X-rays, neutron rays, proton and electron rays), acoustic and ultrasonic detection, electrical and electromagnetic detection, mechanical and optical testing, thermodynamic methods, and chemical analyses [90,91,92,93,94,95,96].

Currently, the main methods applied to non-destructive testing for additive manufacturing, as shown in Table 2, include ultrasonic phased array technology, CT detection technology, and infrared thermographic detection technology. These methods enable the inspection and evaluation of defects in additive manufacturing processes, ensuring the quality and integrity of the produced parts.

### 3.1. Ultrasonic Phased Array Technology

#### 3.1.1. Ultrasonic Phased Array Detection Principle Overview

Ultrasonic phased array technology is based on the principle of Huygens, where multiple independent piezoelectric wafer array elements are used to emit acoustic waves. By controlling the excitation of each array element in a specific sequence, a specific acoustic field is formed, resulting in beam focusing and phase-controlled deflection. When receiving reflected waves, the same method is used to synthesize the signals received by each array element, and the synthesized results are displayed in an appropriate form [97]. The process is shown in Figure 9.

In the process of ultrasonic phased array inspection, the phased array controller triggers high-voltage electric pulses based on the signals transmitted by the ultrasonic detector. Each array element receives an electrical pulse, generating an ultrasonic beam at a specific angle and depth according to the focusing law. When the beam encounters a defect, it is reflected back, and the phased array controller changes the delay time according to the receiving focus law to combine the received signals and form a pulse signal that is then transmitted to the instrument display unit [98].

The unique advantage of ultrasonic phased array technology lies in its continuous and dynamic adjustment of focus size and position, ensuring consistent detection sensitivity and resolution over a wide range without the need for frequent probe replacement. This improves detection efficiency, accuracy, and the real-time and intuitive nature of inspection [99]. The application of ultrasonic phased array inspection technology in additive manufacturing, especially for large and complex structures, is expected to enhance accessibility, applicability, and real-time inspection capabilities [100].

While there have been no specific reports on the application of ultrasonic phased array technology in the inspection of nanocomposite additive manufacturing parts, there have been studies on its application in additive manufacturing molding materials. Further research and development are needed to explore the potential of ultrasonic phased array technology in nanocomposite additive manufacturing inspection.

#### 3.1.2. Application of Ultrasonic Phased Array in the Inspection of Additive Manufacturing Parts

In a study conducted by Han et al. from the Beijing Institute of Aeronautical Manufacturing Engineering, the application of ultrasonic phased array inspection technology in A-100 steel electron beam melt wire forming parts was investigated. The study involved fan sweeping along the deposition direction (Z direction) and perpendicular to it using a 5 MHz one-dimensional line array probe. The results showed that clearer defect signals could be obtained when detecting along the Z direction with an acoustic beam angle of 0°~10°. The clarity of the defect signals gradually weakened and could not be recognized at −30°, and the clarity of the same defect signals differed at different angles. This indicates that the direction and angle of ultrasonic incidence are crucial for identifying microcracks in A-100 steel electron beam wire forming parts. The microstructure of the forming parts also has a significant impact on the selection of the direction and angle of incidence [101] (Figure 10).

While phased array technology saves scanning time and probe adjustment time, it is noted that phased array near-surface clutter is high, leading to more near-surface blind zones and probe partition focusing. Further optimization of phased array inspection technology is necessary. However, phased array inspection technology is considered a future direction for inspection development and is expected to be applied in more parts inspections.

#### 3.1.3. Future Advantages of Ultrasonic Phased Array Full Matrix Focusing Technology

The advantages of ultrasonic phased array full matrix focusing technology include:Simplified setting of detection parameters and operation processes, making it easier to use.The ability to complete multiple detection tasks (multi-angle, multi-focus) with a single probe sweep, improving efficiency.High resolution can be achieved, allowing for detailed defect detection.The detection effect is not affected by the orientation of defects, providing consistent and reliable results.The signal-to-noise ratio is superior to conventional ultrasonic phased array inspection, resulting in clearer and more accurate detection.

To address the effect of additive manufacturing material anisotropy on ultrasonic detection and improve the detection signal-to-noise ratio for small defects, full-matrix focusing imaging can utilize its complete data package and post-processing techniques. By compensating for sound velocity anisotropy and attenuation anisotropy, the signal-to-noise ratio of defect detection can be improved, leading to enhanced accuracy in defect quantification. Further research is needed to explore this aspect in more depth.

### 3.2. CT Detection Technology

#### 3.2.1. CT Introduction

CT inspection technology, also known as computerized tomography inspection technology, involves the reconstruction of a two-dimensional image of a specific level of an object through computer processing. This reconstruction is based on the projection data acquired by penetrating the object with a certain physical quantity, typically X-ray attenuation. The principle of this technology is illustrated in Figure 11. Additionally, CT inspection technology encompasses the construction of a three-dimensional image by employing mathematical methods on a series of two-dimensional images. Importantly, CT inspection technology ensures that the structure of the inspected object remains intact and undamaged [102,103].

In the field of additive manufacturing inspection, CT inspection technology has gained increasing attention due to its advantages. It is not limited by the material and shape of the part, allowing for defect detection as well as internal geometry measurement. International development in CT inspection technology focuses on enhancing equipment system performance, such as improving ray penetration and detection efficiency by using electron linear gas pedal sources or synchrotron radiation sources, reducing source focus size and detector unit size to improve spatial resolution, and transitioning CT systems towards modularization and information technology. CT systems have been applied to inspect the full size range of additive manufacturing products, from macroscopic to microscopic scales. This includes complex fine structure size measurement, deformation evaluation, micron-level defect detection, and modeling of defect morphology and distribution [104,105,106].

**Figure 11 nanomaterials-13-02741-f011:**
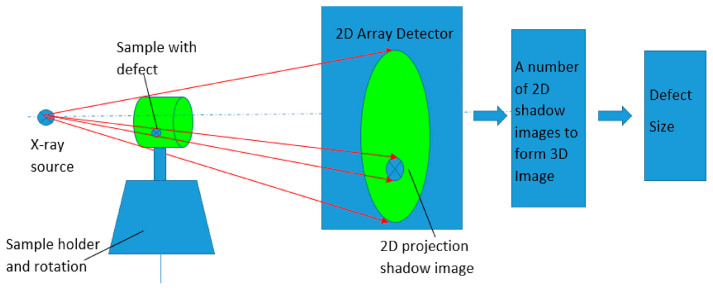
Schematic diagram of an XCT to show its working principle [107].

#### 3.2.2. Application of CT Inspection Technology in Defect Detection

The basic principle of CT defect detection is to use the density difference between the defect and the material to produce the difference in the degree of attenuation of the ray, to show the spatial location of the defect in the CT image and the material, and to produce a distinguishable difference in gray scale, and then realize the recognition of defects within the material. The detection of small defects within a fine structure depends firstly on the ability to obtain defect images with sufficient contrast and resolution.

For CT defect detection technology, the biggest difficulty lies in the detection range and detection accuracy, which are a pair of irreconcilable contradictions. The traditional CT inspection is usually restricted by the location of the region of interest, and the detection process is forced to use the overall cross-section of the component as the scanning area; however, the region of interest only occupies a small area in the scanning area, resulting in a great sacrifice of the detection accuracy, and small-size defects due to the lack of spatial resolution. Small-size defects are difficult to detect due to the lack of spatial resolution. Recently, localized structural CT scanning imaging has become the best solution to this problem. After determining the key position of the component as the detection object, the required spatial resolution is calculated through the defect detection size requirements, then the magnification ratio is adjusted so that the signal generated in the target area during the detection process can be completely received by the detector, and the detection image of the target area is obtained through local reconstruction techniques. Although this method will inevitably introduce a large number of artifacts, if the appropriate scanning parameters are adjusted to increase the contrast difference between the defect and the material, defects such as porosity can still be clearly displayed [108,109,110].

Factors that affect the imaging quality of additive manufacturing defects include part material thickness, structural scattering rays, spatial resolution of the CT system, and density resolution. Thicker part materials lead to a decrease in defect detection capability. The skeletonized structures of additive manufacturing parts can result in more severe scattered rays, affecting the imaging quality of small-size structures near the surface area. Hardware and software processing is required to eliminate the effect of scattered rays on image quality. The performance of the CT system, including spatial resolution and density resolution, is crucial in determining the imaging quality of defects. Spatial resolution affects the recognition of small details and the morphology of defects, while density resolution affects the intensity of image noise at the location of defects.

In a study by Wang et al., X-ray computed tomography (XCT) was used to investigate the internal porosity of additively manufactured parts. XCT was able to quantify the volume of closed and open pores, allowing for a better understanding of pore defects and the quality of the additively manufactured part. XCT data were confirmed through quantitative analysis of both selective laser melting (SLM) for closed pores and binder jetting and polishing (BJP) for open pores. Compared to other testing methods, XCT was found to be an effective method in measuring porosity and providing accurate feedback data for quality control in the additive manufacturing process [107] (Figure 12).

#### 3.2.3. Application of CT Inspection Technology in Molding Size Inspection

Additive manufacturing technology enables the production of complex structural nanocomposites with controllable density, compactness, and multifunctional design. However, the internal contour dimensional inspection of additive manufacturing components, especially those with complex internal structures, poses a challenge. Traditional contact or optical dimensional measurement methods are unable to reach the internal structure surface, making internal contour dimensional inspection a difficult problem for quality control [111,112].

CT dimensional measurement, as a non-contact coordinate measurement technology, is a specific application of CT inspection technology in the field of non-destructive testing (NDT). Compared to traditional coordinate measuring machines (CMM), CT dimensional measurement offers several advantages. It allows for non-destructive testing of geometric quantities on both internal and external surfaces and provides dense point cloud data for quick three-dimensional imaging. Therefore, CT inspection technology is used to achieve high-precision dimensional inspection of the internal contour of additive manufacturing structures.

There are two main difficulties in the current application of CT dimensional measurement. Firstly, the position of small-size structure surfaces in CT images cannot be accurately determined using traditional analytical methods. CT imaging is limited by the system’s performance, and when the structure size is smaller than a specific value, the imaging behavior changes significantly. As a result, the position of the surface of small-size structures shifts from the traditional position based on half-height and width or maximum grayscale gradient, leading to large errors in structure size measurement. Secondly, the influence of additive manufacturing surface roughness on CT dimension measurement is not well understood. The surface roughness of additive manufacturing parts, which is influenced by the forming direction and process, can impact the accuracy of CT dimensional measurements.

In the study by Wu Lei et al., they utilized carbon nanotubes as a photothermal material and sodium citrate particles as a surface distribution hole generator. By incorporating these materials into a homemade UV-curable resin, they created a 3D evaporator using the SLA printing technique. The 3D evaporators were designed to mimic the bionic bird beak and hogwash intestinal wall structures [113] (Figure 13).

To characterize the 3D structural morphology of the printed samples, the dimensions of the sample surfaces were measured using the CT technique. The CT technique proved to be promising for precise dimensional measurements, allowing for the characterization of asymmetric grooves and gradient microcavity arrays on the surface of the 3D evaporators. This suggests that CT can be a valuable tool for accurately assessing the dimensional characteristics of complex additive manufacturing structures.

### 3.3. Infrared Thermal Image Detection Technology

Infrared thermal image inspection is a non-destructive testing method that utilizes the principle of infrared radiation. It involves scanning, recording, or observing the surface of a workpiece to detect defects or internal structural discontinuities based on the changes in the surface temperature field caused by differences in heat transfer. This technology is relatively new compared to traditional testing methods like ultrasound and X-ray, but it offers advantages such as fast detection speed, non-contact operation, non-pollution, intuitive results, and sensitivity to near-surface defects and features. In recent years, infrared thermography has made significant progress and has become an important complement to other nondestructive testing techniques [114].

Infrared thermal imaging inspection techniques can be classified as active or passive, depending on whether human-applied excitation is required. The principle of active infrared thermography is shown in Figure 14. Active infrared thermal imaging uses artificial excitation to induce temperature field changes in the object being inspected, which are then analyzed to obtain internal information. Passive infrared thermal imaging, on the other hand, relies on the temperature field distribution of the object itself for detection and analysis. This can include abnormal heat generation in electric power systems, electronic devices, mechanical parts, or the use of natural conditions such as sunlight or ambient temperature differences [115,116,117].

In the context of additive manufacturing, infrared thermal imaging inspection technology is primarily used for online monitoring of the manufacturing process. It involves monitoring the temperature field and characteristic temperature parameters in real-time to control the process parameters and ensure the stability of the manufacturing process. This can help maintain or improve the quality of the printed parts.

Researchers have also explored the role of in situ infrared thermography in quality control during the additive manufacturing process. They have studied how infrared thermography can be used for closed-loop quality control of powder bed fusion systems and have identified potential defects that may occur during additive manufacturing. They have also investigated the impact of experimental parameters on the quality of additively manufactured parts.

In a study by Abouel Nour et al., optical imaging and infrared thermography were used for the detection of artificially introduced defects in inspected parts. The defects were analyzed through temperature monitoring and thermal image analysis and compared to a baseline to identify and characterize the defects. The study found that the mean temperature of the specimen increased as the number of defects embedded in the part increased. The results demonstrated the feasibility of using thermal imaging systems for defect detection in additive manufacturing [118].

Overall, infrared thermal imaging inspection technology offers valuable capabilities for non-destructive testing and quality control in additive manufacturing processes, allowing for real-time monitoring and control of the manufacturing parameters to ensure the quality and stability of the process [119].

## 4. Discussion and Future Trends

One of the challenges in defect detection for additively manufactured samples is the reliance on traditional 3D image inspection methods, which often require cutting or damaging the samples, resulting in non-real structures being displayed. However, the current state of non-destructive testing (NDT) technology is insufficient to meet the demand for defect detection in nanocomposites. Therefore, we propose several research directions that should be prioritized in the future for NDT in nanocomposite additive manufacturing:Research on the basic problems of the additive manufacturing process: There are still many fundamental issues in the additive manufacturing of nanocomposites that need to be addressed, such as the effects of nanoparticles on light scattering and absorption in SLA processes. Subsequently, it is also necessary to characterize the defects of the products through nondestructive testing, and construct the mapping relationship of process parameters–microstructure–property.Application research of new NDT technology: With the development of nanocomposite additive manufacturing parts in the direction of refinement and complexity, it is difficult to meet the requirements of traditional NDT methods. Therefore, more research should be conducted on the application of new NDT techniques, such as ultrasonic phase control and high-resolution CT.Online inspection method research: Online inspection of additive manufacturing parts is one of the key development directions for the future. Currently, exploratory research has been conducted on online inspection technology of additive manufacturing parts, but there is still a gap from practical application. In-depth research is needed in infrared thermal imaging, optical imaging, laser ultrasound, and other online inspection methods to enable real-time monitoring of products and improve efficiency.Research on online inspection methods: Online inspection of additively manufactured parts is one of the key development directions in the future. At present, exploratory research has been carried out on the online inspection technology of additive manufacturing parts, but there is still a gap from practical application. At present, methods such as infrared detection and other methods to achieve the analysis of nanocomposites such as pores and other defects of its resolution have to be further improved, and, at the same time, it is necessary to conduct in-depth research on infrared thermography, optical imaging, laser ultrasound and other online inspection methods to achieve real-time monitoring of the product and to improve the efficiency of the work.The establishment and improvement of non-destructive testing method standards: Currently, there is no established non-destructive testing standard system for nanocomposite additive manufacturing parts. This lack of standards hinders the wide application of additive manufacturing parts. Therefore, the establishment and improvement of non-destructive testing method standards will also be one of the key development directions in the future.

By addressing these research directions, advancements can be made in NDT for nanocomposite additive manufacturing, enabling the mass application of this technology.

## 5. Summary

The engineering and large-scale application of additively manufactured parts of nanocomposites still face significant challenges and limitations. Internal quality control, internal stress management, and dimensional accuracy evaluation are major obstacles that need to be addressed to promote widespread adoption of additive manufacturing technology. Non-destructive testing (NDT) technology has emerged as a promising approach to address these challenges and improve the additive manufacturing process.

However, a comprehensive review of the current state of NDT technology applied to additive manufacturing reveals that non-destructive testing of nanocomposite parts has not been extensively explored. The focus of research has primarily been on online monitoring using CT technology and infrared imaging. While these techniques have shown potential, they are still in the developmental stages.

To bridge the gap between research and practical application, more comprehensive and advanced NDT methods need to be developed and applied to nanocomposite additive manufacturing. This will require a multidisciplinary approach, incorporating advancements in materials science, imaging technology, data analysis, and machine learning. By addressing the challenges associated with internal quality control, internal stress management, and dimensional accuracy evaluation, additive manufacturing technology can be further optimized for widespread industrial application.

## Figures and Tables

**Figure 1 nanomaterials-13-02741-f001:**
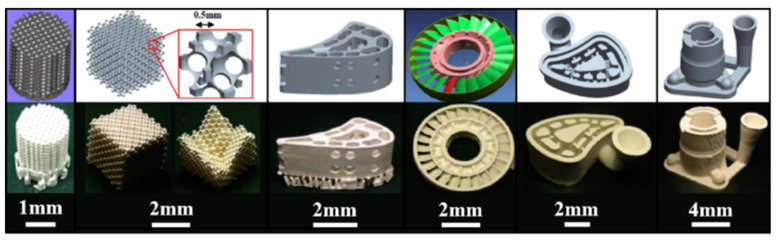
Complex-shaped samples prepared by additive manufacturing methods [11].

**Figure 2 nanomaterials-13-02741-f002:**
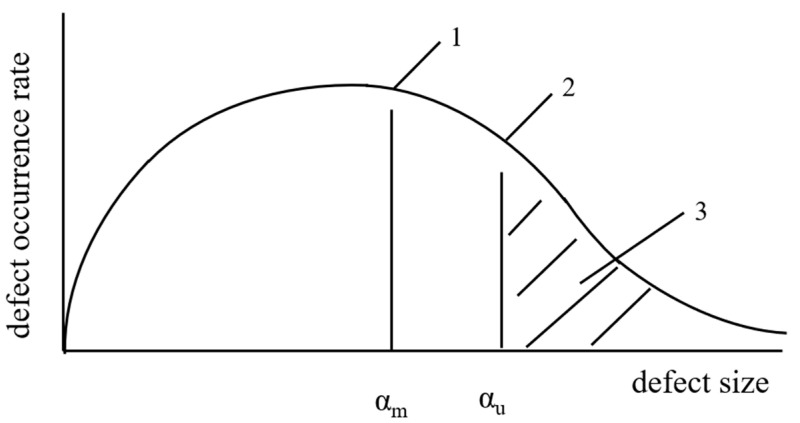
Defect size–occurrence distribution curve. 1. Determined defects to be excluded; 2. Maximum permissible defects determined by stress level; 3. Excluded.

**Figure 3 nanomaterials-13-02741-f003:**
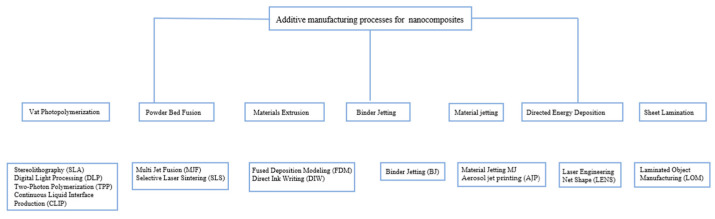
Groups of additive manufacturing technologies by the International Organization for Standardization (ISO)/American Society for Testing Materials (ASTM) [30].

**Figure 4 nanomaterials-13-02741-f004:**
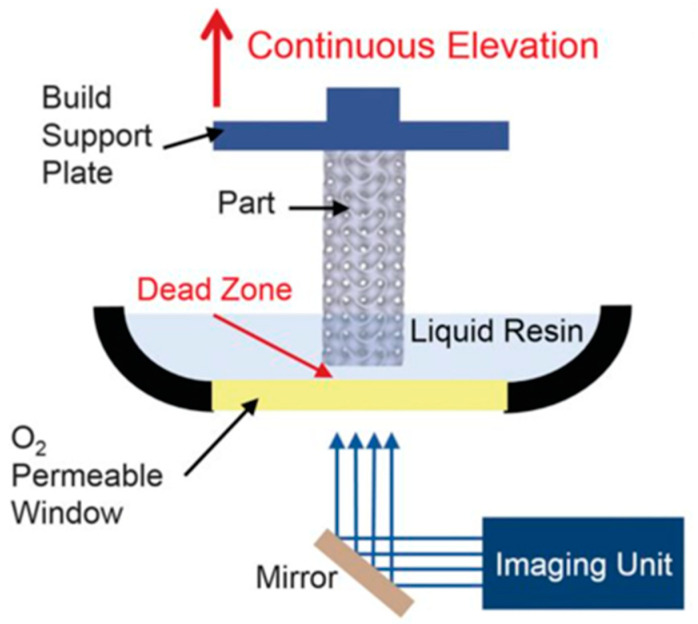
Schematic of CLIP printer [37].

**Figure 5 nanomaterials-13-02741-f005:**
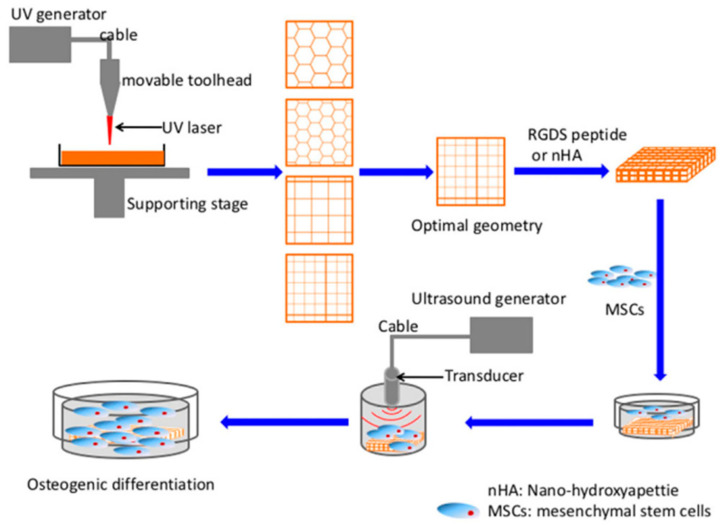
Schematic diagram of 3D printed scaffold fabrication and MSC excitation by low intensity pulsed ultrasound (LIPUS) [38].

**Figure 6 nanomaterials-13-02741-f006:**
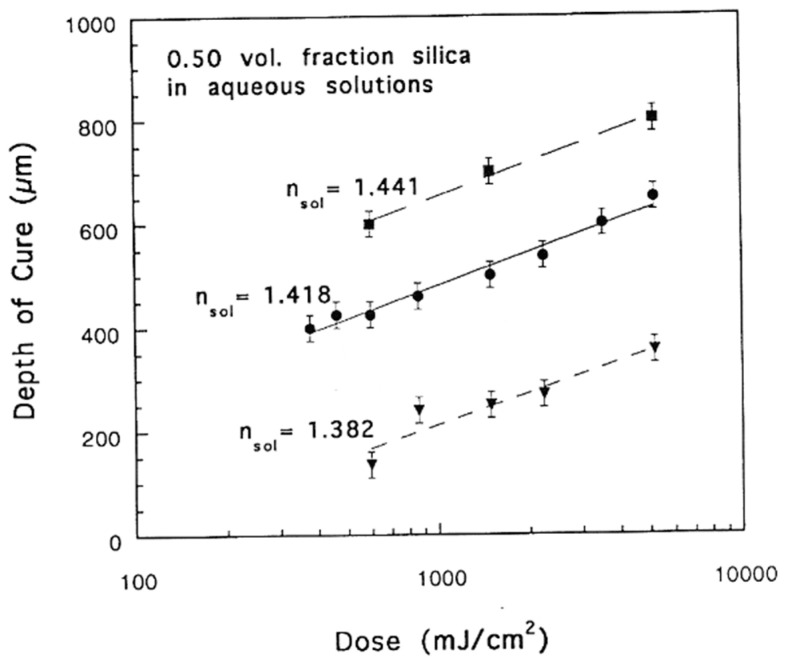
Cure depth versus exposure dose for 0.50 volume fraction silica dispersed in three aqueous UV-curable solutions [40].

**Figure 7 nanomaterials-13-02741-f007:**
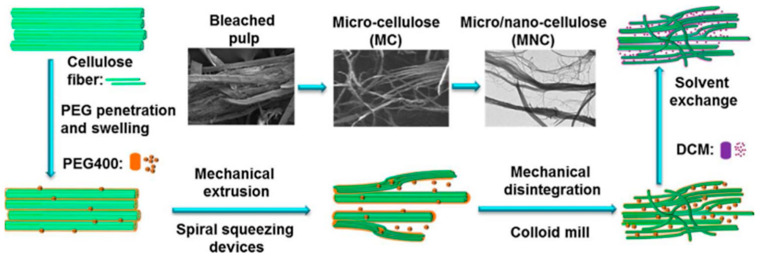
Proposed schematic process of the preparation of micro/nanocellulose (MNC) [56].

**Figure 8 nanomaterials-13-02741-f008:**
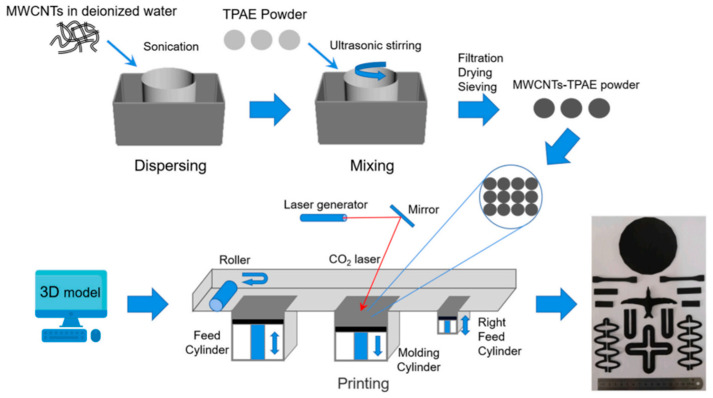
Schematic illustration for preparing route of specific devices [71].

**Figure 9 nanomaterials-13-02741-f009:**
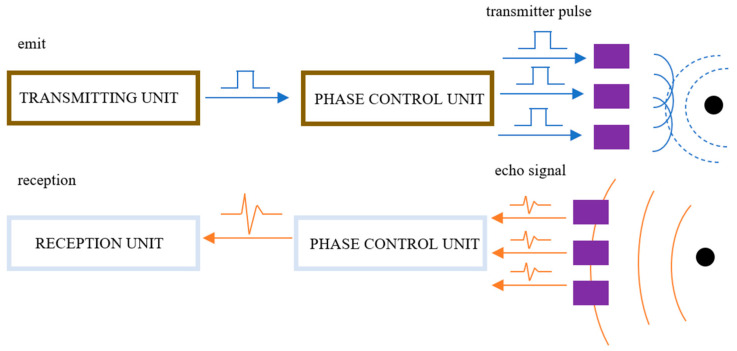
Schematic diagram of ultrasonic phased array transmission, reception, and time delay.

**Figure 10 nanomaterials-13-02741-f010:**

Documentation one-dimensional sector sweep inspection results for 1D line arrays. D_1_, D_2_ is defect [101].

**Figure 12 nanomaterials-13-02741-f012:**
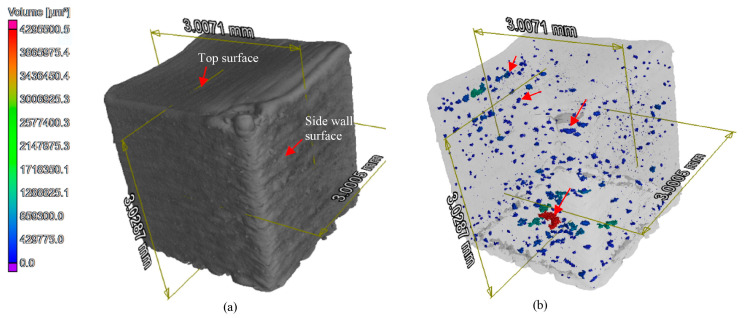
3D XCT images of a cubic sample after reconstruction of 1200 X-ray projections. (**a**) 3D plot of the cubic sample obtained by XCT to show an overall XCT results and (**b**) a transparent 3D plot to those close pores hidden inside the solid cubic sample [107].

**Figure 13 nanomaterials-13-02741-f013:**
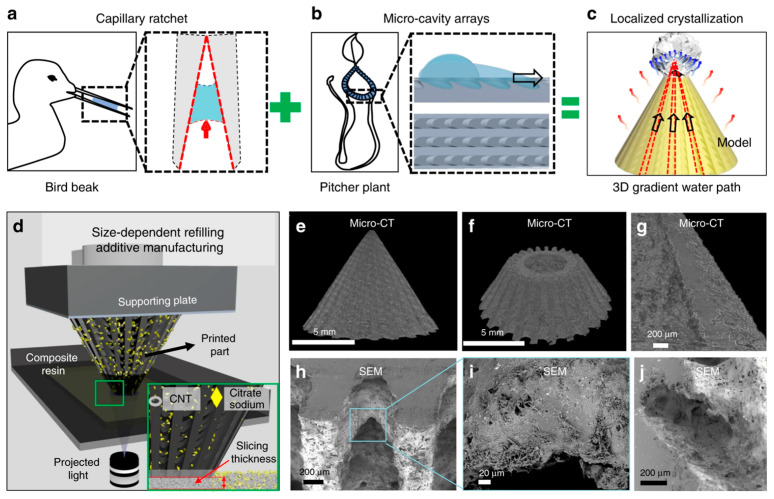
Side view reconstructed Micro-CT image of the biomimetic 3D evaporator. (**a**) The super liquid transportation property of the asymmetric capillary ratchet of the bird beak. (**b**) The super liquid transportation property of the peristome surface of the pitcher plant. (**c**) The inhomogeneous water film induced localized salt crystallization on the biomimetic 3D evaporator and its application in solar-driven water evaporation enhancement. (**d**) Schematic configuration of size-dependent resin refilling induced additive manufacturing based on the continuous DLP 3D printing system. Inset is the scheme of the size-dependent particle refilling process where particles with a dimension larger than the slicing thickness cannot flow along with the refilling resin and are solidified only on the surface of the printed structure. (**e**–**j**) Characterization of the biomimetic 3D evaporator. (**e**) Side view reconstructed Micro-CT image of the biomimetic 3D evaporator [113].

**Figure 14 nanomaterials-13-02741-f014:**
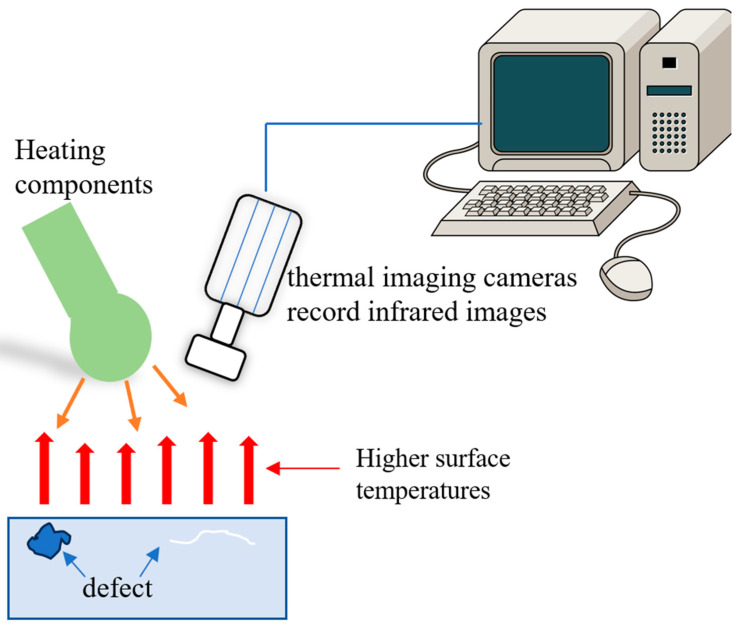
Principle of active infrared thermography.

**Table 1 nanomaterials-13-02741-t001:** Different image analysis tests for additive manufacturing sample.

Imaging Technology	Resolution	Non-Destructive?	Pros and Cons
X-ray Computed Tomography (XRM)	~10 nm	Yes	Non-destructive, macroscopic (closer to real sample information), relatively low resolution
Focused Ion Beam (FIB)	~0.3 nm	No	Destructive (may introduce non-realistic structures), microscopic (poor statistical representation), high resolution
Electron Tomography (ET)	~0.1 nm	No	Destructive (may introduce non-realistic structures), microscopic (poor statistical representation), extremely high resolution
Atom Probe Tomography (APT)	~0.1 nm	No	Destructive (may introduce non-realistic structures), microscopic (poor statistical representation), extremely high resolution
Nuclear Magnetic Resonance Imaging (NMR)	~0.1 mm	Yes	Non-destructive, macroscopic (closer to real sample information), minimum resolution

**Table 2 nanomaterials-13-02741-t002:** Main features of non-destructive testing methods.

Analysis Strategy	Resolution	Advantages	Limitations
ultrasonic testing	0.1–1 mm	Sensitive to defects, fast results, and easy defect localization	Difficult to detect small, thin, and complex parts, need coupling agent coupling, complex shape of the structure is difficult to detect.
ray detection	1 μm	It is not limited by material or geometry and maintains a permanent record. Radiographic inspection is most sensitive to volumetric defects such as porosity	Large investment in equipment; not suitable for on-site online testing, long testing period
infrared detection	0.1–1 °C	Fast, intuitive, accurate. Easy to check the preparation time is short, and check the high efficiency, non-contact. Most cases do not contaminate and do not need to touch the test piece	Low detection depth and low resolution
magnetic particle inspection	0.1–0.5 mm	High sensitivity for testing ferromagnetic materials, easy to operate, reliable results and intuitive display.	Limited to ferromagnetic materials, quantitative determination of defects Difficult to determine depth
osmosis	0.01–0.1 mm	The principle is simple and easy to understand, the equipment is simple, easy to operate, high sensitivity. Intuitive display of defects	Complicated process, can only detect surface opening defects, can not detect the surface of porous materials
eddy current detection	0.1–1 mm	High degree of automation, no need to clean the surface of the specimen, easy to check.	Sensitive to edge effects caused by part geometry, mutations, and prone to false displays

## Data Availability

Not applicable.

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
