# Peer review of "Nano-Additive Manufacturing and Non-Destructive Testing of Nanocomposites"

_nanomaterials, 2023, doi:10.3390/nano13202741_

Round 1

Reviewer 1 Report

The paper is well-organised and found of interest. 

However, the abstract is reads like a summary and the final summary as well both should be revised and so the abstract has to highlight the main findings while the summary has to be a conclusion and emphasise the generic insights gained out of the conducted literature review.

Numerous related work is omitted and in  the manuscript there are some pages with plenty of text with a single or few references at the end of a number of paragraphs. For example from line 62 to line 129 there is no references and same in different places across the manuscript. 

AM is a group of technologies with over 50 different techniques, why only three of them are considered and discussed in this paper. In particular, inkjet printing or material jetting is a well-known technique which is used to print composite uv-curable polymers.

References should come before the full-stop and not after. Please revise.

The recommendations for future work should be strengthen as these are the main findings of this study.

The quality of the figures could be improved.

English could be improved, especially, some unnecessary and basic details have to be removed. 

Reviewer 2 Report

In this review the significant for modern technology topic of detecting defects in 3D printed composite materials containing nano fillers is discussed. In general manuscript is interesting and can be suitable for publication in Nanomaterials. However before publications I suggest to reorganise it slightly according to following comments:

1. In introduction section the brief description of possible defects should be given. Additionally it will be beneficial to highlight briefly what new types of defects can be introduced by nanoparticles. Or what types of "usual" defects (like weak interlayer adhesion) can be enhanced.

2. During reading it seems that review is constructed with two separate parts - 1) methods of 3D printing and 2)methods of detecting of defects. However it is expected from the title that in text the information about special defects produced by nano fillers in the case of different 3D printing approach will be found. This information indeed exist in text but in not obvious manner. I suggest to highlight connections between parts of review by highlighting the following order of discussion in the first part: method of 3D printing->what defects are introduced (enhanced) by nano fillers -> what method of defect detection was used in literature -> what method is better (faster, most simple, most informative, have the highest resolution).

3. It will be easy to understand the review if the methods for detecting of defects will be described prior to describing of defects in different 3D printing processes. Moreover, the most interesting and original part of review is exactly description of methods of detecting defects. The description of 3D printing methods can be easily found elsewhere and are well known. Thus I suggest to move "methods" part on the first place.

4. The better title will be "Non-destructive testing of defects in 3D printed nanocomposites". It is obvious that the text following this title will be a review, thus addition of "a review" in title looks redundant.

Reviewer 3 Report

This review titled "Additive manufacturing and non-destructive testing of nanocomposites: A Review” is devoted to the non-destructive testing and additive technologies dealing with nanocomposites. This is a very actively developing area of research and technology, therefore studies in this area are greatly needed.  

The article can be published in Nanomaterials after a major revision. To publish the article in Nanomaterials, authors should make the following corrections and additions to the review text:

1. It is advisable to reflect in the title specifically the area the review is aimed at, since there have been a lot of review on Additive manufacturing in recent years. Since the authors say in the abstract that the review is aimed at the additive manufacturing process in nano-additive manufacturing, it is advisable to add “nano” in the title.

2. Line 24-25 – authors should add vat polymerization, binder and material jetting, otherwise the total number of basic physical processes is narrowed.

3. There are no references to the Figures in the text. Because of this, it is not very clear what they illustrate and what is the idea of the illustration. Authors should add them.

4. In the chapter related to SLA in the example [31] it is not indicated that this is a nanocomposite and what phase is nanosized in it.

5. In the same chapter some formulas are given, but the dimensions of the quantities are not indicated; they should be added, otherwise it is not clear in which units Сd Cw, E are being measured.

6. In the chapter about FDM, line 259-260 - it is not deciphered what CNT, GNP and PBT are, the same in line 267 about CNF; in general, all abbreviations should be explained when they appear in the text for the first time.

7. It is not clear why the authors did not include the methods of inkjet and aerosol jet printing and also micro plotter printing - currently they are used in the most of works concerning 2D- and 3D-printing of nanoparticles and nanocomposites. I recommend the authors to add a chapter about these methods.

8. Chapter about Non-destructive testing, line 408 - I doubt that chemical analysis is NDM - if the authors meant some kind of non-destructive chemical method, then this needs to be specified.

9. I would like the authors to provide the characteristics of the Ultrasonic Phased Array Detection method in terms of spatial resolution so that its applicability to the study of nanocomposites could be assessed. It would be desirable to state the same thing more clearly for the CT method.

10. Since nanoparticles are smaller in size than the resolution of the detection methods considered in the review, they can introduce defects with sizes that may not be detected by these methods. Therefore, in the chapter “Discussion and future trends” it is necessary to indicate that we need to work to increase the spatial resolution of detection methods without losing the speed of analysis.

Moderate editing of the English language is required as non-standard words are often used, which makes the text difficult to understand.

Round 2

Reviewer 1 Report

The authors have satisfactorily addressed the reviewer comments and the paper has been improved.

Reviewer 2 Report

Thank you for careful consideration of comments. I think that article can be accepted now.

Reviewer 3 Report

Almost all my comments were generally correctly  taken into account, as a result the manuscript was significantly improved and it can be published in the present form.